# Discovery of $\hat{C}_2$ rotation anomaly in topological crystalline insulator SrPb

Wenhui Fan[1,2,8], Simin Nie[3,8], Cuixiang Wang [1,2,8], Binbin Fu[1,2], Changjiang Yi[1], Shunye Gao[1,2], Zhicheng Rao[1,2], Dayu Yan[1,2], Junzhang Ma [4,5], Ming Shi[5], Yaobo Huang [6], Youguo Shi[1,2,7✉], Zhijun Wang [1,2✉], Tian Qian [1,7✉] & Hong Ding [1,2,7]

Topological crystalline insulators (TCIs) are insulating electronic states with nontrivial topology protected by crystalline symmetries. Recently, theory has proposed new classes of TCIs protected by rotation symmetries $\hat{C}_n$, which have surface rotation anomaly evading the fermion doubling theorem, i.e., $n$ instead of $2n$ Dirac cones on the surface preserving the rotation symmetry. Here, we report the first realization of the $\hat{C}_2$ rotation anomaly in a binary compound SrPb. Our first-principles calculations reveal two massless Dirac fermions protected by the combination of time-reversal symmetry $\hat{T}$ and $\hat{C}_{2y}$ on the (010) surface. Using angle-resolved photoemission spectroscopy, we identify two Dirac surface states inside the bulk band gap of SrPb, confirming the $\hat{C}_2$ rotation anomaly in the new classes of TCIs. The findings enrich the classification of topological phases, which pave the way for exploring exotic behavior of the new classes of TCIs.

[1] Beijing National Laboratory for Condensed Matter Physics and Institute of Physics, Chinese Academy of Sciences, Beijing, China. [2] University of Chinese Academy of Sciences, Beijing, China. [3] Department of Materials Science and Engineering, Stanford University, Stanford, CA, USA. [4] Department of Physics, City University of Hong Kong, Kowloon, Hong Kong. [5] Swiss Light Source, Paul Scherrer Institute, Villigen, PSI, Switzerland. [6] Shanghai Synchrotron Radiation Facility, Shanghai Advanced Research Institute, Chinese Academy of Sciences, Shanghai, China. [7] Songshan Lake Materials Laboratory, Dongguan, Guangdong, China. [8] These authors contributed equally: Wenhui Fan, Simin Nie, Cuixiang Wang. ✉email: ygshi@iphy.ac.cn; wzj@iphy.ac.cn; tqian@iphy.ac.cn

Since the advent of time-reversal symmetry $\hat{T}$ protected topological insulators characterized by topological $\mathbb{Z}_2$ invariants[1,2], a generalization of the topology concept to the band insulators is topological crystalline insulators (TCIs)[3], protected by discrete crystalline symmetries. They are of tremendous interest in condensed matter physics and materials science[4–9] in view of the rich variety of crystalline symmetries in solids[10,11]. However, although 230 crystallographic space groups of nonmagnetic materials imply the enormous potential to realize abundant TCIs, theoretical predictions of TCIs are for a long time limited to materials with mirror or glide-mirror symmetries, such as SnTe[12] and KHgSb[13]. The SnTe-based compounds are confirmed to be TCIs by observation of Dirac-cone surface states[14–16], and the experimental signature of "hourglass" surface states in KHgSb has been reported[17].

Recently, mapping the symmetry data of nonmagnetic materials with gapped band structures to the topological invariants has been carried out by several theoretical groups to greatly simplify the process for identifying the topologically nontrivial insulators[18–21], leading to the discovery of a substantial number of TCIs by first-principles high-throughput screening[22–25]. In this elegant strategy, a set of up to six $\mathcal{Z}_{n=2,3,4,6,8,12}$ numbers, also named symmetry-based indicators, has been used to classify the topological states including the new classes of TCIs protected by $n$-fold ($n = 2, 4, 6$) rotational symmetries $\hat{C}_n$ instead of mirror or glide-mirror symmetries[26]. The rotational symmetry-protected TCIs have $n$ unpinned Dirac cones at generic $k$ points (related by $\hat{C}_n$) on the surface preserving the rotational symmetry $\hat{C}_n$, which are connected by $n$ one-dimensional (1D) helical states on the hinges parallel to the rotation axis because of the higher-order bulk-boundary correspondence[27,28]. The number of the surface Dirac cones in the new classes of TCIs is $n$ rather than $2n$ as expected from the well-known fermion multiplication theorem in two-dimensional (2D) time-reversal-invariant systems. The violation of the theorem is a result of the quantum anomaly induced by the breaking of continuous rotation symmetry, which is termed rotation anomaly for short[27].

Owing to the great theoretical efforts, the new classes of TCIs have been predicted in some candidates, including bismuth[29], Bi$_4$Br$_4$[30,31], and Ba$_3$Cd$_2$As$_4$[32] with $\hat{C}_2$ rotation anomaly, anti-perovskite Sr$_3$PbO with $\hat{C}_4$ rotation anomaly[6], and SnTe with $\hat{C}_2$ and $\hat{C}_4$ rotation anomalies on different surfaces[27,28]. However, the experimental evidence of rotation anomalies is very rare except the Dirac cones previously observed on the (001) surface of SnTe[14–16], which are now reinterpreted as the $\hat{C}_4$ rotation anomaly[27,28]. In this work, by combining first-principles calculations and angle-resolved photoemission spectroscopy (ARPES) experiments, we have revealed the $\hat{C}_2$ rotation anomaly on the (010) surface of the binary compound SrPb, where two Dirac-cone surface states protected by the combination of $\hat{T}$ and $\hat{C}_{2y}$ are identified.

## Results

### $\hat{C}_2$ rotation anomaly in SrPb.
SrPb crystallizes in a centrosymmetric orthorhombic unit cell ($a = 5.018$ Å, $b = 12.23$ Å, and $c = 4.648$ Å) with space group $Cmcm$ (SG #63)[33], as shown in Fig. 1a. Both Sr and Pb atoms occupy the $4c$ $(0, y, 1/4)$ Wyckoff position with the internal parameter $y = 0.132$ and $0.422$ for Sr and Pb, respectively. The lattice is formed by a stacking of slightly buckled SrPb layers along the [010] direction. The point group of the structure is $D_{2h}$ generated by inversion $\hat{I}$, twofold rotational symmetries ($\hat{C}_{2x}$ and $\{\hat{C}_{2y}|00\frac{1}{2}\}$), screw rotational symmetry $\{\hat{C}_{2z}|00\frac{1}{2}\}$, mirror symmetries ($\hat{M}_x$ and $\{\hat{M}_z|00\frac{1}{2}\}$), and glide-mirror symmetry $\{\hat{g}_y|00\frac{1}{2}\}$. $\{\hat{C}_{2n}|ijk\}$ is the twofold rotational

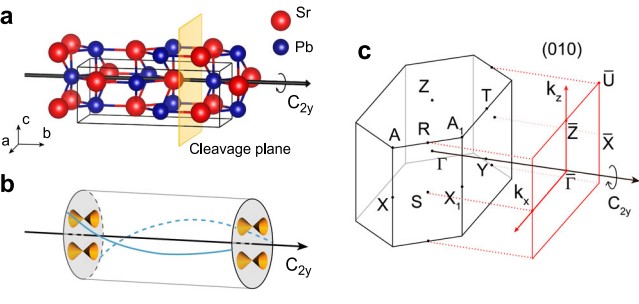

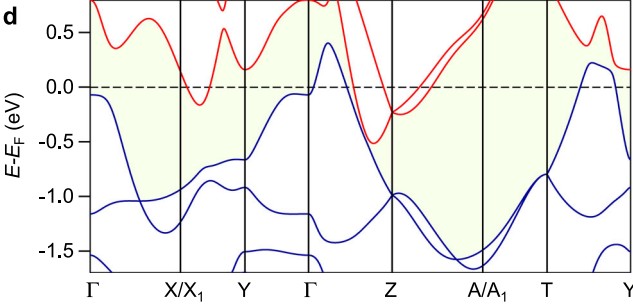

**Fig. 1 Crystal structure and band structure of SrPb. a** Crystal structure of SrPb with the rotation axis $\hat{C}_{2y}$ along the $b$ axis. The yellow plane indicates the cleavage position at the (010) surface. **b** Schematic of topological surface states with $\hat{C}_2$ rotation anomaly, showing two Dirac cones on each surface perpendicular to the $\hat{C}_2$ rotation axis, which are connected by two helical edge modes on the side surface. **c** Bulk BZ and (010) surface BZ of SrPb with high-symmetry points indicated. **d** Calculated bulk bands of SrPb along high-symmetry lines with SOC. The blue and red curves represent valence and conduction bands, respectively. The shadow region indicates the continuous gap between the valence and conduction bands.

symmetry with respect to the vector $\boldsymbol{n}$ followed by a fractional lattice translation $i\boldsymbol{a} + j\boldsymbol{b} + k\boldsymbol{c}$, and $\{\hat{M}_{\boldsymbol{n}}|ijk\}$ is the mirror reflection with respect to the plane with the normal vector $\boldsymbol{n}$ followed by a fractional lattice translation $i\boldsymbol{a} + j\boldsymbol{b} + k\boldsymbol{c}$, where $\boldsymbol{a}$, $\boldsymbol{b}$, and $\boldsymbol{c}$ are three unit cell vectors.

The full-potential linearized augmented-plane-waves method as implemented in the WIEN2K code is employed to calculate the band structure of SrPb (see Methods for the calculation details), which is shown in Fig. 1d. It is clear that the band-gap opening due to spin-orbit coupling (SOC) occurs around the Fermi level ($E_F$), leading to the existence of a continuous direct gap between the conduction and valence bands at each $k$ point in the full Brillouin zone (BZ). Therefore, the Fu-Kane topological invariants $\mathbb{Z}_2$ ($\nu_0; \nu_1\nu_2\nu_3$)[34] and symmetry-based indicators $\mathcal{Z}_{2,2,2,4}$ ($z_{2,i=1,2,3}; z_4$)[18,19] for band insulators are well defined for SrPb in SG #63 (note that $\nu_i = z_{2,i}$ with $i = 1, 2, 3$). Because of the existence of inversion symmetry in SrPb, the Fu-Kane parity criterion[34] can be used to easily calculate the topological invariants $\mathbb{Z}_2 = (\nu_0; \nu_1\nu_2\nu_3)$, where $\nu_0$ ($\nu_{i=1,2,3}$) is the strong (weak) topological index. Based on the parities of occupied bands at eight time-reversal-invariant momenta (TRIM) listed in Table 1, the Fu-Kane topological invariants $\mathbb{Z}_2$ are computed to be (0;110), which are consistent with the previous calculations[22,23]. The symmetry-based indicators $\mathcal{Z}_{2,2,2,4}$ are computed to be (1102)[22,35], with $\mathcal{Z}_4 = 2$ indicating the existence of higher-order topology in the TCI SrPb.

Given that the mapping from symmetry-based indicators to topological invariants is one-to-many mapping (it is a one-to-four mapping for SrPb), the mirror Chern numbers ($n_{\hat{M}_x}$ and $n_{\hat{M}_z}$) are calculated in order to confirm the exact topological

phase in SrPb. In the presence of time-reversal symmetry, the Chern number ($n_{\hat{M}}$) can be identified in half of the plane by the 1D Wilson loops of the two mirror eigenvalues ($+i$ and $-i$) subspaces[36]. The Wannier charge centers of the $k_z$-directed and $k_x$-directed Wilson loops as a function of $k_y$ for the $k_x = 0$ and $k_z = 0$ planes are shown in Fig. 2a, b, respectively. The results show that $n_{\hat{M}_x}$ of the $k_x = 0$ plane and $n_{\hat{M}_z}$ of the $k_z = 0$ plane are 0 and $-2$, respectively, signaling the existence of topologically protected surface states on the surfaces preserving the mirror symmetry $\hat{M}_z$. According to ref. [19], we conclude that SrPb, with the indicators (1102) and invariants $n_{\hat{M}_x} = 0$, $n_{\hat{M}_z} = 2$, belong to the TCI phase in Table 2, which has nontrivial values of the $\hat{C}_2$ anomaly indices. The invariant $\nu_{\hat{C}_{2y}} = 1$ guarantees the $\hat{C}_{2y}$ rotation anomaly in the (010) surface of SrPb. Two Dirac-cone surface states can be stabilized by $(\hat{T} \cdot \hat{C}_{2y})^2 = 1$ and protected from annihilation by the coexistence of $\hat{T}$ and $\hat{C}_{2y}$.

**Surface band calculation.** The nonzero topological invariants suggest the existence of topologically nontrivial surface states, which are calculated with the Green's function methodology, as shown in Fig. 2c, d. We find two type-II surface Dirac cones on the (010) surface, which are consistent with the nontrivial topological invariant $\nu_{\hat{C}_{2y}} = 1$. Interestingly, the Dirac cones are

**Table 1 Parities of four pairs of occupied bands at eight TRIM.**

| TRIMs | Position | Parities order | Parity product |
|---|---|---|---|
| 1Γ | $(0, 0, 0)$ | ++−− | + |
| 1Y | $(0, 2\pi, 0)$ | ++−− | + |
| 2S | $(-\pi, \pi, 0)$ $(\pi, \pi, 0)$ | +−−− | − |
| 1Z | $(0, 0, \pi)$ | ++−− | + |
| 1T | $(0, 2\pi, 0)$ | ++−− | + |
| 2R | $(-\pi, \pi, \pi)$ $(\pi, \pi, \pi)$ | ++−− | + |

The positions of the TRIM are given in units of $(\frac{1}{a}, \frac{1}{b}, \frac{1}{c})$.

constrained on the high-symmetry line $\bar{X} - \bar{\Gamma}$ on account of the nonzero $n_{\hat{M}_z}$ Chern number. Next, we study the influence of the breaking of mirror symmetry $\hat{M}_z$ on the two surface Dirac cones. After applying a tiny shear strain (about 2%), the third primitive lattice vector $\vec{c}$ becomes $(0.1, 0, c)$ in unit of Å, while the other primitive lattice vectors and the internal parameters of Sr and Pb atoms remain unchanged, resulting in a crystal without $\hat{M}_z$ but preserving $\hat{C}_{2y}$. The results of surface state calculations are shown in Fig. 2e, f, which show that the two surface Dirac cones cannot be gapped out but move away from the high-symmetry line $\bar{X} - \bar{\Gamma}$ to generic $k$ points due to $(\hat{T} \cdot \hat{C}_{2y})^2 = 1$. In view of the higher-order bulk-boundary correspondence, the higher-order TI with two helical hinge states is expected on the hinges parallel to the $y$ axis, as shown in Fig. 1b.

**Bulk electronic structures of SrPb.** To examine our theoretical prediction of the nontrivial topological phase, we have carried out systematic ARPES measurements on the (010) cleavage surface of SrPb single crystals. We determine momentum locations perpendicular to the sample surface by photon energy ($h\nu$) dependent ARPES measurements (see Supplementary Fig. 1 in Supplementary Information). Figure 3c, d shows the in-plane Fermi surfaces (FSs) measured with $h\nu = 58$ and 34 eV, which are consistent with the calculated FSs in the $k_y = 0$ (Fig. 3e) and $k_y = 2\pi/b$ (Fig. 3f) planes, respectively. As seen in Fig. 3a, the calculations exhibit quasi-2D FSs extending along the $k_y$ direction and three-dimensional (3D) FSs located around the $k_y = 0$ and $k_y = 2\pi/b$ planes in the 3D BZ. The quasi-2D FSs have two semicircular contours in the $k_y = 2\pi/b$ plane (Fig. 3d, f) and they merge to form a nearly elliptical contour enclosing the $\Gamma$ point in the $k_y = 0$ plane (Fig. 3c, e). In addition, Fig. 3c shows two faint semicircular contours near the $\Gamma$ point, which are ascribed to momentum broadening of the FSs near the Y point because of short mean free path of the photoelectrons. On the other hand, the 3D FSs are observed near the Z point in the $k_y = 0$ plane (Fig. 3c, e) and on the $Y - X_1$ line in the $k_y = 2\pi/b$ plane (Fig. 3d, f).

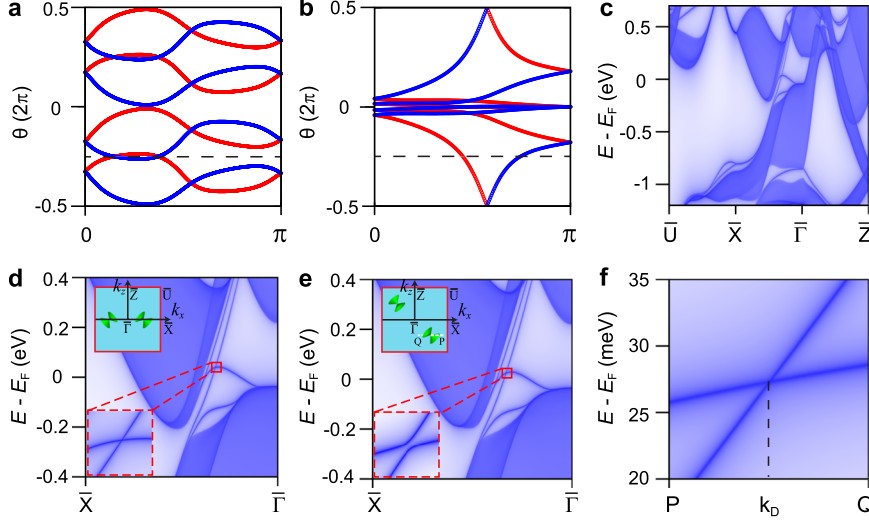

**Fig. 2 WCCs and Dirac surface states of SrPb. a, b** WCCs of the $k_z$-directed and $k_x$-directed Wilson loops as a function of $k_y$ in the $k_x = 0$ and $k_z = 0$ planes, respectively. The red and blue curves represent the flow of the WCCs for the bands with mirror $+i$ and $-i$ eigenvalues, respectively. The horizontal dashed lines are reference lines. **c, d** Surface band structures of SrPb on the (010) surface. **e** Surface Dirac cone on the line $\bar{X} - \bar{\Gamma}$ is gapped out when the $\hat{M}_z$ is broken. The upper insets in (**d, e**) schematically show the positions of the type-II Dirac surface cones in the (010) surface BZ. The lower inset in (**e**) clearly shows the gapped Dirac cone. **f** Dispersions of the $\hat{C}_{2y}$ rotational symmetry-protected surface Dirac cone along the line $P - Q$ on the (010) surface. The positions of $P$, $k_D$, and $Q$ are $(0.2077, -0.0022)$, $(0.2052, -0.0022)$, and $(0.2027, -0.0022)$, respectively. The positions are given in unit of Å$^{-1}$.

| $Z_{2,2,2,4}$ | $(\nu_1\nu_2\nu_3)$ | $\{\hat{M}_z\|00\frac{1}{2}\}$ | $\hat{M}_x$ | $\{\hat{g}_y\|00\frac{1}{2}\}$ | $\{\hat{C}_{2y}\|00\frac{1}{2}\}$ | $\{\hat{C}_{2x}\}$ | $\hat{I}$ | $\{\hat{C}_{2z}\|00\frac{1}{2}\}$ |
|---|---|---|---|---|---|---|---|---|
| 1102 | 110 | −2 | 0 | 0 | 1 | 1 | 1 | 0 |

**Table 2 Symmetry-based indicators and topological invariants of SrPb.**

$n_{\hat{M}_z}$ of the $k_z = \pi/c$ plane is 0 and not shown here.

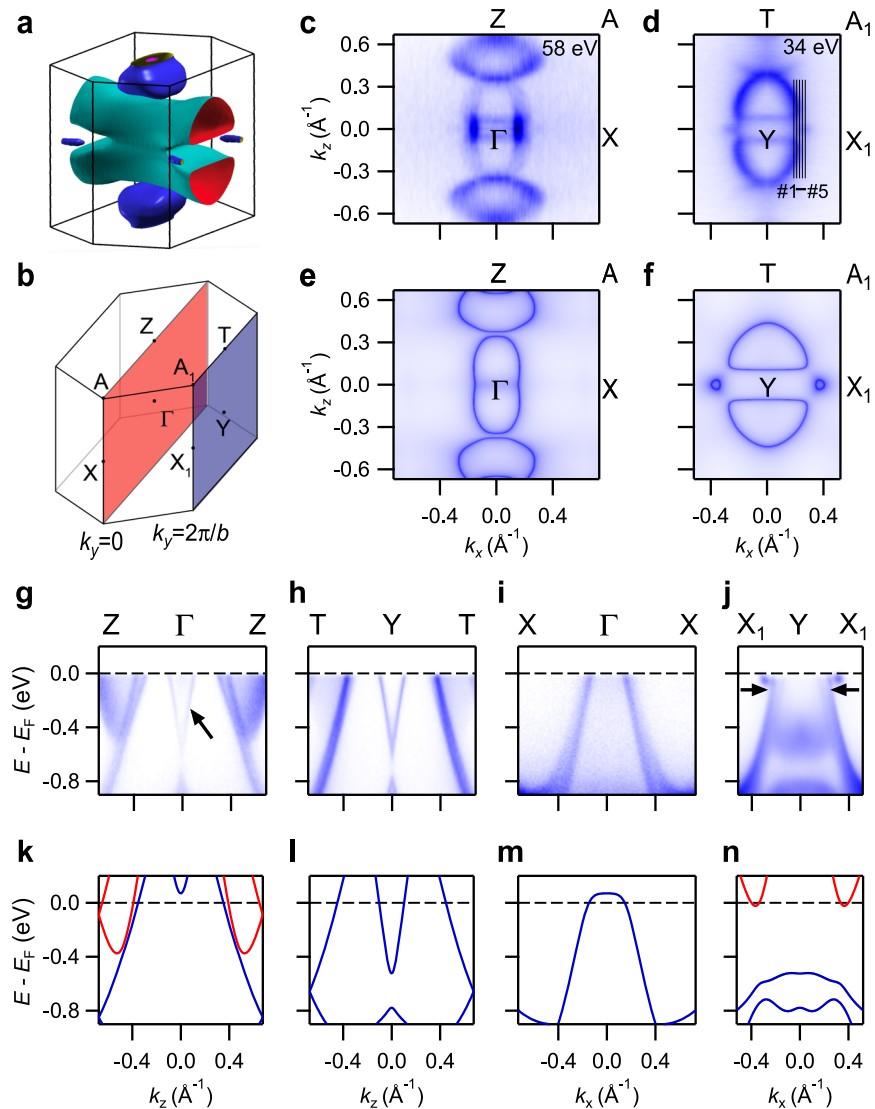

**Fig. 3 Bulk electronic structures of SrPb. a** Calculated FSs of bulk states in the 3D BZ. **b** 3D bulk BZ with the $k_y = 0$ and $k_y = 2\pi/b$ planes marked with red and blue colors, respectively. **c, d** ARPES intensity plots at $E_F$ recorded on the (010) surface with $h\nu = 58$ and 34 eV, respectively. The data in (**c, d**) are symmetrized with respect to $\Gamma - X$ and $Y - X_1$, respectively. The black lines in (**d**) indicate momentum locations of the cuts in Fig. 4i, j. **e, f** Calculated FSs in the $k_y = 0$ and $k_y = 2\pi/b$ planes, respectively. **g-j** ARPES intensity plots showing band dispersions along $\Gamma - Z$, $Y - T$, $\Gamma - X$, and $Y - X_1$, respectively. The arrows in (**g**) and (**j**) indicate extra bands compared with the calculated bulk bands. **k-n** Calculated bulk bands along $\Gamma - Z$, $Y - T$, $\Gamma - X$, and $Y - X_1$, respectively. We note that the Fermi level in the calculations is shifted down by 0.14 eV to match the experimental results because the samples are slightly hole-doped.

**Dirac surface states protected by $\hat{C}_2$ rotational symmetry.** In Fig. 3g–n, we compare the experimental bands with the calculated bulk bands along the high-symmetry lines $\Gamma - Z$, $Y - T$, $\Gamma - X$, and $Y - X_1$. As shown in Fig. 1c, both $\Gamma - Z$ and $Y - T$ are projected onto $\bar{\Gamma} - \bar{Z}$ while both $\Gamma - X$ and $Y - X_1$ are projected onto $\bar{\Gamma} - \bar{X}$ on the (010) surface. The experimental bands are generally consistent with the calculated bulk bands. In

addition, we identify some extra bands, as indicated with arrows in Fig. 3g, j. The extra band dispersion at the $\Gamma$ point in Fig. 3g is identical with the one at the $Y$ point in Fig. 3h. As the extra band is very sharp, we attribute it to 2D surface states rather than the momentum broadening of 3D bulk states, the latter usually smears the band dispersions. The surface band is located at the lower boundary of the valence band continuum and has nothing

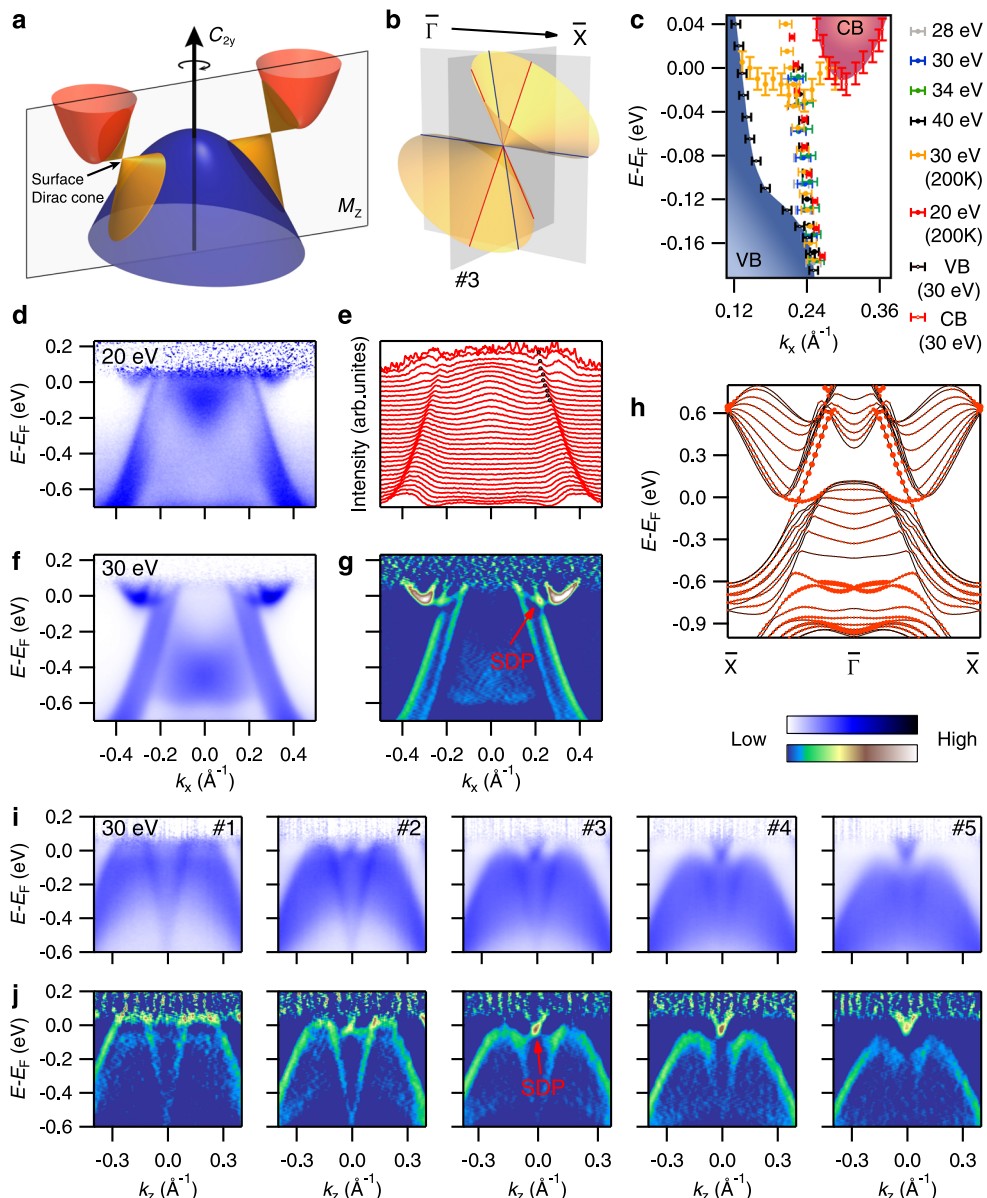

**Fig. 4 Dirac surface states on the (010) surface. a** Schematic band structures of SrPb showing two surface Dirac cones protected by $\hat{C}_{2y}$ rotational symmetry. **b** Schematic of tilted surface Dirac cones, showing dispersions of Dirac-like band crossings along and perpendicular to the tilt direction. **c** Near-$E_F$ band dispersions along $\bar{\Gamma} - \bar{X}$ extracted from the ARPES data collected at different photon energies and temperatures. **d** ARPES intensity plot along $\bar{\Gamma} - \bar{X}$ measured with $h\nu = 20$ eV at 200 K. **e** Momentum distribution curves of the data in (**d**). **f** ARPES intensity plot along $\bar{\Gamma} - \bar{X}$ measured with $h\nu = 30$ eV at 200 K. **g** 2D curvature intensity plot of the data in (**f**), showing two surface bands cross each other forming a surface Dirac point (SDP). The method of 2D curvature was developed in[45]. **h** Calculated band structures along $\bar{\Gamma} - \bar{X}$ of a slab with a thickness of five unit cells along the [010] direction for termination A, whose cleavage position is indicated as a yellow plane in Fig. 1a. The size of red dots scales the contribution from the outmost two SrPb layers. The Fermi level of the calculated bands is shifted down by 0.14 eV. **i, j** ARPES intensity plots and corresponding 2D curvature intensity plots along the cuts perpendicular to $\bar{\Gamma} - \bar{X}$, whose momentum locations are indicated in Fig. 3d. The data in (**i**) were collected with $h\nu = 30$ eV at 200 K. The ARPES data in Fig. 4 are divided by the Fermi-Dirac distribution function.

to do with the topology of the band gap between the valence and conduction states. On the other hand, we observe two hole-like bands across $E_F$ along $\Gamma - X$ and $Y - X_1$ (Fig. 3i, j), whereas the calculations (Fig. 3m, n) indicate only one hole-like bulk band. Our ARPES measurements show that the inner band dispersion exhibits obvious variation while the outer one remains unchanged with varying photon energy (see Supplementary Fig. 2 in Supplementary Information). The inner band is identified as the valence band of bulk states, while the outer one is attributed to the surface band, which is located in the band gap between the valence and conduction bulk states.

We also discern the sign of a nearly flat band at $E_F$ in the bulk band gap from the ARPES data collected at 30 K (see Supplementary Fig. 2 in Supplementary Information). In order to exhibit the near-$E_F$ band more clearly, we have carried out ARPES measurements at 200 K, which reduce the influence of the Fermi cutoff effect so that the bands above $E_F$ can be observed. The ARPES data collected at 200 K in Fig. 4f, g reveal a relatively flat band near $E_F$, which emanates from the bottom of the conduction band and extends to the valence band. Remarkably, the two bands in the bulk band gap forming a crossing at ~20 meV below $E_F$. In Fig. 4c, we plot the band dispersions extracted

from the experimental data collected along $\bar{\Gamma} - \bar{X}$. The observation of Dirac-like band crossing in the bulk band gap is in accord with our theoretical expectation, whereas the experimental surface band dispersions in Fig. 4c are obviously different from the tight-binding calculations in Fig. 2d, which do not consider the real situation of cleavage surface. We thus performed slab calculations considering two kinds of terminations (see Supplementary Fig. 4 in Supplementary Information). While the calculations for both terminations show Dirac-like surface states along $\bar{\Gamma} - \bar{X}$, the calculated surface bands for termination A in Fig. 4h well reproduce the experimental results. The excellent consistency between experiment and calculation provides compelling evidence for the nontrivial topology of the bulk band gap.

We further investigate band dispersions of the Dirac surface states perpendicular to $\bar{\Gamma} - \bar{X}$. Figure 4i, j shows band dispersions measured along a series of cuts perpendicular to $\bar{\Gamma} - \bar{X}$, whose momentum locations are indicated in Fig. 3d. A Dirac-like band crossing is clearly resolved at ~20 meV below $E_F$ along cut #3. These experimental data demonstrate that the Dirac-cone surface states are strongly tilted along the $\bar{\Gamma} - \bar{X}$ direction as the schematic drawing in Fig. 4b. As seen in Fig. 3d, the equal-energy contour of the Dirac-cone surface states is not a closed FS but an open arc connecting to the FSs of bulk states. The tilted Dirac-cone surface states are reminiscent of the type-II Weyl and Dirac semimetals[37,38].

## Discussion

By combining theoretical calculations and ARPES experiments, we have demonstrated the $\hat{C}_2$ rotation anomaly in SrPb. We draw a schematic plot of the band structures of SrPb in Fig. 4a, which shows a pair of surface Dirac cones protected by $\hat{C}_{2y}$ rotational symmetry in the bulk band gap. The Dirac points lie on the high-symmetry line $\bar{\Gamma} - \bar{X}$ due to the constraint of mirror symmetry, which is similar to the case in SnTe[12,14–16]. As has been demonstrated in the SnTe-based compounds, the Dirac points can be tuned to move along the $\bar{\Gamma} - \bar{X}$ line by element substitution with the mirror symmetry being preserved[39]. If the mirror symmetry is broken while the rotational symmetry kept by a particular lattice distortion, the Dirac points will move away from the $\bar{\Gamma} - \bar{X}$ line. In addition to 2D Dirac surface states at generic locations in momentum space, rotational symmetry-protected TCIs also have 1D hinge states at generic locations in real space, which have been predicted to have a quantized conductance of $ne^2/h$[27]. It is highly desirable to explore these unique properties of the new classes of TCIs in the future.

## Methods

**Sample synthesis**. Single crystals of SrPb are grown using self-flux methods. The starting materials Sr (distilled dendritic pieces, 99.8%) and Pb (shot, 99.999%) are mixed with a molar stoichiometric ratio of 1:1.32 in a glove box filled with high-purity Ar gas. The mixture is placed in an alumina crucible and sealed in an evacuated quartz tube. The tube is heated to 1000 °C and kept for 10 h. The sample is quickly cooled down to 830 °C and then slowly cooled down to 730 °C at a rate of 2 °C/h followed by centrifuging to remove the excess of Pb. Finally, the shiny crystals SrPb are obtained on the bottom of the crucible. Because the crystals are extremely sensitive to air and water, it is important to store the crystals in an Argon atmosphere.

**Angle-resolved photoemission spectroscopy**. All the ARPES data shown are recorded at the "Dreamline" beamline of the Shanghai Synchrotron Radiation Facility. We conducted complementary experiments at the Surface and Interface Spectroscopy beamline at the Swiss Light Source, the CASSIOPEE beamline of Soleil Synchrotron, and beamline 13U of National Synchrotron Radiation Laboratory. The energy and angular resolutions are set to 15–30 meV and 0.2°, respectively. All the samples for ARPES measurements are mounted in a BIP Argon (>99.9999%)—filled glove box, cleaved in situ, and measured at 25 K and 200 K in a vacuum better than $5 \times 10^{-11}$ torr.

**Calculation method**. The calculations are performed using the full-potential linearized augmented-plane-waves method as implemented in the WIEN2K code[40]. The exchange and correlation potential is treated within generalized gradient approximation of Perdew, Burke and Ernzerhof type[41]. SOC is included in the calculation as a perturbative step. The atomic radii of the muffin-tin sphere RMT are 2.5 bohrs for Sr and Pb. The $9 \times 9 \times 10$ kmesh is used in the BZ. The s orbitals of Sr and p orbitals of Pb are used to construct the maximally localized Wannier functions[42], which are then used to calculate the boundary states by an iterative method[43,44].

## Data availability

Materials and additional data related to this paper are available from the authors upon request.

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

## Acknowledgements

We acknowledge Yuting Qian, Hongming Weng, and Chen Fang for fruitful discussions. We acknowledge Chunxiao Liu, Shiming Zhou, and Jie Zeng for their help in determining the crystal structure of SrPb samples by single-crystal XRD measurements. This work was supported by the Beijing Natural Science Foundation (Z180008), the Ministry of Science and Technology of China (2016YFA0401000, 2016YFA0300600, 2017YFA0403401 and 2017YFA0302901), the National Natural Science Foundation of China (U1832202, U2032204, 11974395, 11888101, and 11622435), the Chinese Academy of Sciences (QYZDB-SSW-SLH043, XDB33000000, and XDB28000000), the Beijing Municipal Science and Technology Commission (Z171100002017018 and Z181100004218005), the Center for Materials Genome, the NCCR-MARVEL funded by the Swiss National Science Foundation, the Users with Excellence Program of Hefei Science Center CAS (2019HSC-UE001), and the K.C. Wong Education Foundation (GJTD-2018-01).

## Author contributions

T.Q. and Z.W. supervised the project; W.F., B.F., and T.Q. performed the ARPES measurements with the assistance of S.G., Z.R., J.M., M.S., Y.H., and H.D.; S.N. and Z.W. performed ab initio calculations; C.W., C.Y., D.Y., and Y.S. synthesized the single crystals; S.N., W.F., T.Q., and Z.W. analysed the experimental and calculated data, plotted the figures, and wrote the paper; All authors participated in discussions and modifications of the paper.

## Competing interests

The authors declare no competing interests.
