## [Peer Review File · Nature Communications]

REVIEWER COMMENTS

Reviewer #1 (Remarks to the Author):

This work reports the experimental discovery of a C2 protected TCI, SrPb. This TCI is characterized by two surface Dirac points, which cannot be realized in a standalone 2D system. Though such a C2-TCI was proposed in the Bi (ref.29), it was not confirmed in the experiment so far. The present work is the first to discover C2-TCI. Thus, I believe this will be an important work in studying new TCI materials. The theory and experiments are well explained. The manuscript is well organized. I would recommend the publication of this work only after the following questions are addressed.

The surface state calculations are based on the tight-binding parameters extracted from the bulk calculations. Although the TB calculations should present the same topology as the ab initio, they commonly provide different surface dispersions. When comparing to the experiment, the ab initio calculations seem to be necessary to avoid any misinterpretation. The authors must know it from the TaAs surface states (e.g., Phys. Rev. X 5, 031013, 2015). Their theoretical surface states in Fig. 2d have some differences from the experiment results in Fig. 4c. What are the atomic terminations in calculations and experiments? In the experiment, surface states below and above the Dirac cones are missing. I suggest checking with ab initio methods and different atomic terminations.

Reviewer #2:

This paper presents both first-principle calculations and ARPES that SrPb is a topological crystalline insulator protected by time-reversal symmetry and \hat{C}_{2y} rotational symmetry. Based on the results of first-principles calculations predicting Dirac cone surface states satisfying \hat{C}_{2y} rotation symmetry in the 2D BZ for (010) surfaces, the authors observe topological Dirac bands satisfying \hat{C}_{2y} rotation symmetry in the $\bar{\Gamma} - \bar{X}$ line by ARPES. In particular, the ARPES data provide a detailed assignment of the bulk/surface bands through a 3D momentum-resolved observation using VUV incident photon energy dependence, and the data at 200 K clearly extract the characteristics of the Dirac dispersion, which is globally convincing.

I recognize that this manuscript provides new experimental finding in topological materials, which can be definitely a solid piece of new information in this field. On the other hand, we would like to make the following suggestions for a better manuscript.

General comment

I personally think impact of this finding is less clear, at least with current written version since plenty of arguments exist on symmetry protection of topological states in recent topological materials search. Discussion on *novel macroscopic* property/functionality in SrPb is required to put authors finding significant. Probably related to this, the authors have discussed tunable band structure in SrPb as “If the mirror symmetry is broken while the rotational symmetry is kept by a particular lattice distortion, the surface Dirac points will move to the generic momenta away from the $\bar{\Gamma} - \bar{X}$ line”. In Sn-Te based compound, tunable band nature related to characteristic crystalline symmetry has been already demonstrated in experiments [for example, PRB 87, 155105 (2013)].

Another comments

1. To make findings further solid, spin-polarized arpes experiment is preferred.
2. Is there any particular reason why the authors use 2D curvature did not choose the 2nd. derivative commonly used in conventional method? While this is minor comment, reader may have similar question.

~~~~~

## To Referee #1

~~~~~

This work reports the experimental discovery of a C_2 protected TCI, SrPb. This TCI is characterized by two surface Dirac points, which cannot be realized in a standalone 2D system. Though such a C_2 -TCI was proposed in the Bi (ref.29), it was not confirmed in the experiment so far. The present work is the first to discover C_2 -TCI. Thus, I believe this will be an important work in studying new TCI materials. The theory and experiments are well explained. The manuscript is well organized. I would recommend the publication of this work only after the following questions are addressed.

The surface state calculations are based on the tight-binding parameters extracted from the bulk calculations. Although the TB calculations should present the same topology as the *ab initio*, they commonly provide different surface dispersions. When comparing to the experiment, the *ab initio* calculations seem to be necessary to avoid any misinterpretation. The authors must know it from the TaAs surface states (e.g., Phys. Rev. X 5, 031013, 2015). Their theoretical surface states in Fig. 2d have some differences from the experiment results in Fig. 4c. What are the atomic terminations in calculations and experiments? In the experiment, surface states below and above the Dirac cones are missing. I suggest checking with *ab initio* methods and different atomic terminations.

We are very grateful to Referee #1 for his/her high evaluation and recommendation of our work. Following Referee #1's suggestions, we have conducted slab calculations with different terminations at the (010) surface. We consider two possible cleavage positions, as indicated in Fig. R1a,b, which produce two kinds of terminations with different dangling bonds. For termination A, the cleaving breaks one chemical bond for each Sr or Pb atom, while for termination B, the cleaving breaks two chemical bonds for each Sr or Pb atom.

The calculated bands for terminations A and B are plotted in Fig. R1c,d, in which the size of red and blue dots scales the contribution of the outmost two SrPb layers. The calculated surface states for termination A in Fig. R1c well reproduce the experimental results in Fig. 4c, while the calculated results for termination B in Fig. R1d are inconsistent. Therefore, we infer that the real cleavage surface should be

termination A. We thank again for Referee #1's valuable suggestions.

We have added the calculated results for termination A to Fig. 4h and indicated the cleavage position of termination A in Fig. 1a. The calculated results for both terminations are added to Fig. S4 in the Supplementary Materials.

Fig. R1 a,b Crystal structure of SrPb with different cleavage positions, indicated as yellow planes, which produce two kinds of terminations. c,d Calculated band structures of slabs with a thickness of five unit cells along the [010] direction for different terminations.

~~~~~

## To Referee #2

~~~~~

This paper presents both first-principle calculations and ARPES that SrPb is a topological crystalline insulator protected by time-reversal symmetry and \hat{C}_{2y} rotational symmetry. Based on the results of first principles calculations predicting Dirac cone surface states satisfying \hat{C}_{2y} rotation symmetry in the 2D BZ for (010) surfaces, the authors observe topological Dirac bands satisfying \hat{C}_{2y} rotation symmetry in the $\bar{\Gamma} - \bar{X}$ line by ARPES. In particular, the ARPES data provide a detailed assignment of the bulk/surface bands through a 3D momentum-resolved observation using VUV incident photon energy dependence, and the data at 200 K clearly extract the characteristics of the Dirac dispersion, which is globally convincing. I recognize that this manuscript provides new experimental finding in topological materials, which can be definitely a solid piece of new information in this field. On the other hand, we would like to make the following suggestions for a better manuscript.

General comment

I personally think impact of this finding is less clear, at least with current written version since plenty of arguments exist on symmetry protection of topological states in recent topological materials search. Discussion on novel macroscopic property/functionality in SrPb is required to put authors finding significant. Probably related to this, the authors have discussed tuneable band structure in SrPb as “If the mirror symmetry is broken while the rotational symmetry is kept by a particular lattice distortion, the surface Dirac points will move to the generic momenta away from the $\bar{\Gamma} - \bar{X}$ line”. In Sn-Te based compound, tuneable band nature related to characteristic crystalline symmetry has been already demonstrated in experiments [for example, PRB 87, 155105 (2013)].

We are very grateful to Referee #2 for his/her affirmation of our work and valuable suggestions to improve our manuscript. In the previous version, we have proposed that the band structure of SrPb can be tuned by breaking the mirror symmetry, for which the surface Dirac points move away from the $\bar{\Gamma} - \bar{X}$ line. As exemplified by Referee #2, tuneable band nature has been already demonstrated in Sn-Te based compounds when the mirror symmetry is preserved, for which the

surface Dirac points move along the $\bar{\Gamma} - \bar{X}$ line. We have incorporated Referee #2's suggestion into the discussion on page 8. We thank again for Referee #2's valuable suggestions.

Another comments

1. To make findings further solid, spin-polarized ARPES experiment is preferred.

We agree that it is desirable to measure the spin polarization of surface bands by means of spin-resolved ARPES. To observe the surface states of SrPb, which lie around the Fermi level, ARPES measurements have to be performed at high temperatures. We found that the lifetime of the electronic states of SrPb was short at high temperatures. Since spin-resolved spectra have much lower signals compared with spin-integrated spectra, it will take much longer time in spin-resolved ARPES measurements. The lifetime is not enough to collect high signal-to-noise ratio data to clarify the spin polarization of surface bands. We hope that the Referee understands the difficulty.

2. Is there any particular reason why the authors use 2D curvature did not choose the 2nd. derivative commonly used in conventional method? While this is minor comment, reader may have similar question.

The second derivative of EDCs or MDCs is commonly used in analysis of ARPES data. In our previous work [1], we developed a new method based on the mathematical concept of curvature, which can improve the visualization of band dispersions compared with the second derivative in many cases. As seen in Fig. R2, the Dirac band dispersions are better visualized in the 2D curvature intensity plot. So, we use the 2D curvature instead of the second derivative in Fig. 4. To make it clear to readers, we cite the paper as Ref. [39] in the figure caption of Fig. 4 in the revised version. We thank Referee #2 for this comment.

[1] P. Zhang, P. Richard, T. Qian, Y.-M. Xu, X. Dai, and H. Ding, Review of Scientific Instruments 82, 043712 (2011).

Fig. R2 **a** surface Dirac point along the cut perpendicular to $\bar{\Gamma} - \bar{X}$. **b,c** corresponding second derivative intensity plots with respect to energy and momentum, respectively. **d** 2D curvature intensity plot.

REVIEWERS' COMMENTS

Reviewer #1 (Remarks to the Author):

My comments are properly addressed by the authors. I suggest the publication of this manuscript now.

Reviewer #2 (Remarks to the Author):

Thank you for considering my comments. I understand spin resolved ARPES is time consuming and it should not be asked necessary component to be published. The revised manuscript is improved, and I recommend for publication.

~~~~~

**To Referee #1**

~~~~~

My comments are properly addressed by the authors. I suggest the publication of this manuscript now.

Thanks for Referee #1's review and comments/suggestions, the slab calculation he/she mentioned increase the credibility of conclusion. Finally, thanks for Referee #1's approval of our work.

~~~~~

**To Referee #2**

~~~~~

Thank you for considering my comments. I understand spin resolved ARPES is time consuming and it should not be asked necessary component to be published. The revised manuscript is improved, and I recommend for publication.

Thanks for Referee #2's review and comments which are very helpful for the improvement of our manuscript. We are grateful to Referee #2 for his/her understanding of difficulty of spin-resolved ARPES measurement. Thanks again for his/her recommendation of our work.